# Hydrogel Coating versus Calcium Sulphate Beads as a Local Antibiotic Carrier for Debridement Procedures in Acute Periprosthetic Joint Infection: A Preliminary Study

**DOI:** 10.3390/gels9090758

**Published:** 2023-09-18

**Authors:** Daniele De Meo, Paolo Martini, Maria Francesca Pennarola, Giovanni Guarascio, Marco Rivano Capparuccia, Giancarlo Iaiani, Vittorio Candela, Stefano Gumina, Ciro Villani

**Affiliations:** 1Department of Anatomical, Histological, Forensic Medicine and Orthopaedics Sciences, Sapienza University of Rome, 00100 Rome, Italy; paolo.martini@uniroma1.it (P.M.); mf.pennarola@gmail.com (M.F.P.); giovanni.guarascio@uniroma1.it (G.G.); vittorio.candela@uniroma1.it (V.C.); stefano.gumina@uniroma1.it (S.G.); ciro.villani@uniroma1.it (C.V.); 2M.I.T.O. (Malattie Infettive in Traumatologia e Ortopedia-Infections in Traumatology and Orthopedics Surgery) Study Group, Policlinico Umberto I Hospital, Viale del Policlinico 155, 00161 Rome, Italy; marco.rivanocapparuccia@uniroma1.it (M.R.C.); giancarlo.iaiani@uniroma1.it (G.I.); 3Department of Public Health and Infectious Diseases, Sapienza University of Rome, 00100 Rome, Italy

**Keywords:** hydrogel, calcium sulphate beads, local antibiotics, periprosthetic joint infection, debridement, DAPRI, DACRI

## Abstract

Periprosthetic joint infections (PJI) are among the most difficult complications to treat in orthopaedic surgery. Debridement, antibiotics, and implant retention (DAIR) represent an efficient strategy for acute PJI, especially when resorbable local antibiotic carriers and coatings are used. The aim of this pilot study was to evaluate the difference between using antibiotic-loaded hydrogel (ALH) and calcium sulphate (CS) beads in the DAIR procedure. We analysed 16 patients who had been treated since 2018 for acute PJI, namely eight patients with knee PJI (50%), seven with hip PJI (43.7%), and one with shoulder PJI (6.2%). Nine patients were treated with the Debridement, Antibiotic Coating and Retention of the Implant (DACRI) method, while seven were treated with the Debridement, Antibiotic Pearls, Retention of the Implant (DAPRI) method. We found no significant differences between the two groups in terms of age, sex, the American Society of Anesthesiologists risk score, Charlson Comorbidity Index, localisation, days from onset to diagnosis and pathogenesis. Furthermore, no differences were found between the DACRI and DAPRI groups in terms of infection control (15 patients, 93.75% with *p* = 0.36) and last C-Reactive Protein values (*p* = 0.26), with a mean follow-up of 26.1 ± 7.7 months. Treatment for one patient affected by knee Candida albicans PJI in the DACRI group was not successful. In conclusion, DAPRI and DACRI appear to be safe and effective treatments for PJIs. This evidence will encourage the development of new clinical research into local carriers and coatings for use in acute implant-associated infections.

## 1. Introduction

Periprosthetic joint infection (PJI) is the most devastating complication in joint replacement surgery, burdened by a high comorbidity and mortality rate [1,2]. Biofilm plays a pivotal role in PJI pathogenesis and can be defined as a community of micro-organisms embedded in a complex matrix of extracellular polymeric substances characterised by their strong attachment to inorganic materials (i.e., joint replacement implants) [3]. Biofilm formation has been described as a time-dependent process divided into stages of attachment, proliferation, maturation, and emigration. The structure of mature biofilms limits the penetration of antibiotics by preventing infection eradication [4]. Antibiotic and surgical treatments are combined with to eradicate the biofilm or, in cases of acute infections, prevent it from reaching full maturity. Acute PJI can be subdivided into hematogenous or postoperative categories [5,6]. Current management strategies for acute PJI include systemic antibiotic-targeted treatments and a Debridement, Antibiotics, and Implant Retention (DAIR) surgical strategy. This technique is considered bone stock-sparing and less invasive, with acceptable success rates varying between 41 and 75% [7,8]. The efficacy of DAIR is higher in postoperative series than in hematogenous ones [9]. In both cases, optimal results are achieved when there is a shorter time between the onset of symptoms and the surgical procedure [10]. Several DAIR techniques have been presented, especially for hip and knee PJIs [11,12]. Currently, DAIR results have improved by combining several intraoperative actions and tools. One of the most interesting tools available to surgeons is resorbable local antibiotic carriers and coatings [13], which hold the advantage of a local high concentration without systemic side effects. Among the carriers available in the literature, a calcium sulphate matrix loaded with an antibiotic of choice is the most tested in vitro and in vivo [14] and the most published in PJI settings [15]. The elution of antibiotic agents from calcium sulphate beads lasts for several weeks while they are progressively adsorbed [16]. The combination of DAIR and antibiotic-loaded calcium sulphate (CS) beads, which took the name ofDAPRI procedure (Debridement, Antibiotic Pearls, Retention of the Implant) has been widely discussed as a means to enhance acute PJI treatment [17] with positive results.

Antibiotic-loaded hydrogels (ALH) are applied differently than other carrier. Hydrogels are used as a tool to prevent biofilm adhesion in newly implanted hardware with a burst release of the surgeon’s chosen antibiotic, which undergoes complete reabsorption via hydrolytic degradation within 72 h and completely releases the antibiotic contained within it [18]. It is mainly used for prophylaxis [19,20]. Clinical use in PJI involves adding a targeted local antibiotic within the hydrogel during one- or two-stage exchanges, with encouraging results [21,22,23]. Table 1 summarises the differences in use and composition between the two antimicrobial carriers. In a previous article, we described a surgical technique in which we combined a DAIR procedure with ALH, but no updated clinical data have ever been presented [24]. 

These local carriers could be used as an additional weapon in treating acute PJI alongside the standard DAIR procedure. The purpose of this study was to conduct preliminary research to assess whether applying ALH in a DAIR procedure is as effective as CS beads application in treating acute PJI in a single-centre experience. The author’s hypothesis was that local antibiotic delivery was clinically effective regardless of the type of carrier used if targeted to the pathogens.

## 2. Results and Discussion

### 2.1. Population

Among all PJIs treated since 2018, we evaluated 16 cases that fulfilled the inclusion criteria, with a mean age of 67.3 ± 10.8 years. Eight of the patients were female (50%). The mean Charlson Comorbidity Index (CCI) was 3.1 ± 2.8 (range 0–10), and the mean American Society of Anesthesiologists (ASA) risk score was 2.2 ± 0.4. Eight patients had knee PJI (50%), seven were affected by hip PJI (43.7%), and one had shoulder PJI (6.2%). Eight PJI occurred postoperatively (50%), and eight were defined as hematogenous late acute infections. The mean time between the onset of infection and diagnosis was 22.1 ± 9.1 days. Four patients had polymicrobial infections (25%), three of which had postoperative PJI. *Staphylococcus aureus* was the main pathogen (seven patients, 43.7%) followed by *Escherichia coli* and *Pseudomonas aeruginosa* (three patients for each pathogen, 18.7%). One patient had culture-negative PJI (6.2%). Patients were divided into two groups: the first group was treated with DACRI (nine patients, 56.3%) and the second group was treated with DAPRI (seven patients, 43.7%). There was no significant difference between age, sex, ASA, CCI, localization, or days from onset to diagnosis and aetiology. Table 2 presents detailed data about the whole population. 

### 2.2. Treatment and Outcomes

Nine patients were treated with DACRI and seven with DAPRI (Table 3). When feasible, local antibiotics were targeted to the pathogen. Dual antibiotics were applied locally in the vast majority of cases (13 patients, 81.2%). The most common combination was gentamicin and vancomycin (10 patients, 62.5%). In the DACRI group, a single antibiotic was used in three cases: the first two cases had a monomicrobial *Streptococcus dysgalactiae* PJI and the third had a polymicrobial PJI (*coagulase negative staphylococci*—*CoNS* and *Streptococcus dysgalactiae*) isolated preoperatively. In both cases, ALH loaded with vancomycin was used. The third case was a *Candida albicans* PJI in which ALH was loaded with fluconazole. The mean length of stay was 23 days, in which systemic antibiotic therapy was administered via IV. After discharge, oral antibiotic therapy was administered for 12 weeks postoperatively.

After a mean follow-up of 26.1 ± 22.2 months, infection control with no need for further antibiotic therapy was achieved in fifteen patients (93.8%). One patient underwent septic revision surgery in the DACRI group. The patient was affected by a knee *Candida albicans* PJI that relapsed after antifungal therapy discontinuation. The patient achieved infection control after a two-stage exchange followed by antifungal suppressive therapy prolonged one year after surgery. The mean C-reactive protein (CRP) at the last follow-up was 3.7 ± 3.7 mg/L. No differences were found between the DACRI and DAPRI groups in terms of infection control (*p* = 0.3624) and CRP values (*p* = 0.2680). A statistically significant difference was observed in terms of follow-up between DACRI and DAPRI (40.4 ± 19.8 and 7.7 ± 1.8 months, respectively) with *p* = 0.0007. Table 4 presents detailed data about the patients involved in the current study.

### 2.3. Discussion

Debridement, irrigation, local antibiotic delivery, retention of fixed implants, and mobile parts exchange in a setting of a dedicated MDT approach achieved high rates of infection control in acute PJI in this series (93.8%). The DAIR procedure without antibiotic-loaded carriers appeared to have lower rates of infection control [25], but the literature is difficult to compare considering the multiple factors that impact DAIR success and the significant differences among surgical and antibiotic protocols. As previously mentioned, several preoperative factors play a pivotal role in treating acute PJIs. Kunutsor et al. showed the impact of geographical location, baseline age, type of infection and localizations in a meta-analysis of all-joints DAIR [26]. As for factors that may impact control rates, a few characteristics were highlighted: the amount of saline lavage, typically up to 6–9 L, is a well-established protocol and frequently reported in the literature, but no real evidence-based data are available [10]. Similar low-quality evidence was presented for changing drapes and surgical setup between the “dirty” and “clean” parts of the procedure. Theoretically, this operation could decrease contamination of the surgical site [27]. The exchange of modular components is another intraoperative measure that theoretically reduces biofilm presence and improves debridement of the intraarticular space, especially in knee PJIs, where polyethylene removal allows access to the posterior capsule [28]. A multicentric study demonstrated a failure reduction of 33% when modular components were changed [29]. Methylene blue dye can bind to eukaryotic cells and bacterial biofilms. Furthermore, *Staphylococcus epidermidis* has demonstrated successful staining of biofilms over implants in vitro [30]. Its efficacy has also been described in clinical applications, but further research is needed [31]. In the authors’ opinion, an intraarticular methylene blue dye injection allows better visualization of the intraarticular space and its debridement. In fact, we have used all these aforementioned procedures in our debridement protocol.

Since its proposal, the DAPRI procedure has improved the results achieved by the DAIR procedure by up to 77.5% of infection-free patients at a 2-year minimum follow-up [17]. Indelli et al.’s procedure involved using CS beads loaded with targeted antibiotics, but it is not the only improvement they described. In their DAPRI procedure, they performed chemical, mechanical, and thermally-guided biofilm removal on the retained components [17]. As for mechanical debridement, they used a 2% Chlorhexidine gluconate-added brush to scrub all the visible surfaces of the retained implants, also supported by other authors [32]. As for thermally-guided debridement, the authors used an argon beam coagulator on the retained implant surface as it is believed to detach biofilm from the implant. None of these techniques were used in this series. As for chemically removing the biofilm, a commercially available acetic acid, benzalkonium chloride based surgical lavage solution, from the previously described DAPRI procedure was used [17]. In the present study, a combination of povidone iodine, hydrogen peroxide, and saline sterile solution was routinely applied to the surgical site after a radical debridement of all the intra-articular synovial and reactive fibrous tissue. The proportions used were adopted by Balato et al. [33]. CS beads are local carriers that treat residual bacteria left in the joint and prevent colonization of newly exchanged mobile parts. Its ability to deliver high concentrations of antibiotics locally for weeks is also appealing. 

From the authors’ perspective, its long-lasting activity is not of any further benefit in a debridement procedure setting. After surgery, residual bacteria are exposed to an initial high burst of antibiotic release that decreases for days until its complete resorption. In the immediate aftermath of surgery, hematoma formation requires protection against bacteria colonization and biofilm persistence. After the first few days, what is not already defeated by local and systemic combinations at their highest concentrations cannot be further defeated. 

Bearing in mind this rationale, ALH plays a role in this particular treatment. The present study highlights how the DACRI procedure is non-inferior to the DAPRI procedure in this small series of patients. ALH protects the surfaces of the implant from recolonization and acts as a local burst release of antibiotics during the first few days. Locally administered doses of antibiotics differed significantly when we used DAC hydrogel, specifically 5 cc of hydrogel combined with 125 mg of vancomycin and 100 mg of gentamicin. When Stimulan was used, we added 5 cc of CS beads containing 500 mg of vancomycin and 120 mg of gentamicin. With the same amount of carrier, lower doses of antibiotic could be delivered with ALH but with a completely sudden release. CS beads require more time for resorption. For CS beads, the release rate is size-dependent since small beads resorb faster than large ones. Regarding limitations associated with using CS beads, i.e., hypercalcemia, heterotopic ossifications and prolonged wound discharge [15], none were found in our series. From the authors’ perspective, the placement of large or medium CS beads must be limited to the subfascial planes or in the joint, with a watertight closure. There are several differences between ALH and CS beads; therefore, no direct comparison could be made between them. Nevertheless, the DACRI and DAPRI procedures presented in this study are both effective at combining local and systemic delivery. We believe that local antibiotics may be useful in the first hour after surgery, which can be achieved with both ALH and CS beads. Although ALH has never been reported in the DAIR procedure, some data are available on one- and two-stage exchanges for chronic PJI [21,22,23,34]. In a retrospective case-control study, Zagra et al. described its use in two-stage exchange cementless revisions in hip PJIs, using the ALH during the reimplantation stage on 27 patients compared to the control group. No infections were found in the ALH group, whereas four infections were found in the control group (*p* = 0.11) [21]. Franceschini et al. found two early failures in a series of 28 hip PJIs that underwent a two-stage exchange with cementless components coated with ALH [34].

Clinical results of ALH applied to uncemented hip one-stage revision for PJIs were described in another case series of ten patients, where there was no infection recurrence after 3.1 years of follow-up [23]. Another series compared two different approaches: 22 PJIs were treated with cementless one-stage exchange coated with ALH and compared to 22 PJIs were treated with cementless two-stage exchange without coating. No differences in terms of infection recurrence were found [22]. 

The present study had several limitations, mostly due to its retrospective nature and not having used randomization to divide the two groups of treated subjects. Another limitation was the small sample size; however, it should be considered that this sample is homogeneous and treated with a systematic multidisciplinary team (MDT) approach. Another limitation was the length of the follow-up, which was done for a minimum of six months and turned out to be a variable with statistically significant differences between the DACRI and DAPRI groups. Debridement usually fails for recurrences [35], and even if the authors considered six months of follow-up sufficient to detect the onset of an acute infection recurrence, it may not be enough time to exclude low-grade infection recurrences.

## 3. Conclusions

In conclusion, DAPRI and DACRI appear to be safe and effective treatments for PJIs. The evidence reported here encourages the authors to continue studying hydrogel application as an antibiotic local carrier in acute PJI. A prospective randomized trial is currently under evaluation by our research group. This evidence could also stimulate research in other fields of acute implant-associated infections, such as acute fracture-related infections, where local antibiotic strategies are currently under investigation. A similar study about ALH and CS beads in surgically treating acute and chronic fracture-related infections is already in progress at our institution. Nevertheless, randomized clinical trials with a longer follow-up and a greater sample size are necessary to confirm the data that emerged from our study.

## 4. Materials and Methods

### 4.1. Study Design

We conducted a retrospective single-centre study by reviewing the electronic medical records of our hospital for patients admitted with a diagnosis of acute PJI. The PJI definition was confirmed according to EBJIS criteria [36]. PJIs were defined as acute postoperative infections occurring within six weeks after index surgery. PJIs were defined as hematogenous if the diagnoses were made six weeks after the onset of symptoms. The criteria included the presence of an acute PJI, DAIR procedure associated with antibiotic-loaded CS beads or ALH, and the absence of multiple localised infections. The exclusion criteria included a DAIR procedure performed without local carriers or coatings, DAIR performed for chronic infections, and patients with an end-stage cancer diagnosis (prognosis < 6 months). Preoperative data on sex, age, BMI, ASA risk score, and CCI were reported. Intraoperative surgical strategy, microbiology, type of systemic and local antibiotic, and duration of antibiotic therapy were recorded. As for follow-up, electronic files from our outpatient clinic were screened to assess the last CRP available, the length of follow-up, and the outcome of infection, which was defined in the 2018 International Consensus Meeting criteria for infection control [37]. This study was conducted in compliance with the ethical principles of the Declaration of Helsinki. Informed consent regarding the collection and analysis of surgery-related data was obtained from all participants included in the study. For the purpose of this study, there was no direct contact with patients. 

### 4.2. Perioperative Treatment and Surgical Strategy

All patients were treated by the same MDT focusing on bone and joint infections. In a multidisciplinary meeting, we discussed treatment strategies on a case-by-case basis and decided to combine them with local and systemic antibiotic therapy. All patients underwent one or multiple attempts of preoperative joint aspiration performed by orthopaedic surgeons where synovial fluid culture was sent in blood culture vials to the microbiology laboratory. Leucocyte count and PMN percentage were investigated in all cases. Whenever feasible, alpha-defensin and leukocyte esterase were performed. Based on the joint aspiration results, antibiotic local treatment was discussed and decided on in the MDT meeting. 

Preoperative antibiotic treatment was prescribed only if sepsis was present. In the other cases, patients started systemic therapy immediately before surgery. A total of 1 g of tranexamic acid was administered before surgery followed by another 1 g during surgery, immediately after new mobile components were implanted. Total knee debridement was conducted using a tourniquet. All surgeries were conducted by the first and senior authors (DDM and CV) with the same surgical equipment and the same debridement and irrigation protocol, beginning with previous scar excision, especially in postoperative cases. A methylene blue dye injection was performed before capsulotomy directly into the joint space to facilitate debridement [27]. If a sinus tract was present, a methylene blue dye injection was performed to guide the fistulectomy. Bone-implant interface and synovial multiple biopsies were performed (four to six) for microbiological cultures, which were brought to a microbiology laboratory and cultured for at least fourteen days. Solid samples were analysed by culture for common germs and liquid samples were seeded in liquid culture media (BD BACTEC). Germ identification was conducted by our microbiology team using the MALDI-TOF system and antibiograms according to EUCAST. Histology biopsies were also performed. Mobile components were explanted and sonicated. After a thorough assessment of the implant stability, debridement was completed, and low-pressure pulse lavage was performed with a saline solution up to 9 L. An antiseptic solution was placed according to Balato et al. [33]. New sterile equipment and sterile fields were prepared, and the antiseptic solution was irrigated again. Two surgical strategies were performed: 

***DACRI procedure***: In a group of patients, new mobile components and retained implant surfaces were coated with antibiotic-loaded hydrogel (Defensive Antibacterial Coating, DAC—Novagenit, Mezzolombardo, Italy) before joint capsule closure. For every procedure, if vancomycin (250 mg) or gentamicin (100 mg) + vancomycin (125 mg) was used, 5 cc of hydrogel was used. If a combination of vancomycin and meropenem was used, 2 vials (5 cc each) containing one antibiotic each were prepared and then mixed together before coating. We also applied 250 mg of vancomycin and 250 mg of meropenem. The technique and the hydrogel used were described in a previous publication [24].

***DAPRI procedure***: In the other group of patients, new mobile components were inserted, and local antibiotic delivery was applied with the aforementioned technique described by Indelli et al. using 4.8 mm CS beads (Stimulan, Biocomposites, Keele, UK) [17]. If a combination of gentamicin and vancomycin was applied, 5 cc was used (500 mg of vancomycin and 120 mg of gentamicin). If vancomycin + meropenem was selected, a combination of two separate preparations (5 cc with 500 mg of vancomycin and 5 cc with 500 mg of meropenem) was then mixed together. Whenever possible, a 5 cc bag was preferred to prevent the risk of prolonged wound discharge, as described in the literature [15].

From 2018 to 2021, the antibiotic ALH was the only antibiotic carrier available in the hospital. Then, antibiotic-loaded CS beads became available in 2022, which we have since used routinely.

In both cases, the joint capsule was closed in a water-tight fashion and wound closure was achieved with closed incisional negative pressure wound therapy.

### 4.3. Statistical Analysis

Statistical analysis was performed with R version 3.4.4 (R Core Team (2018); R: A language and environment for statistical computing; R Foundation for Statistical Computing, Vienna, Austria). Categorical data were expressed as a percentage while continuous variables were reported as mean and range. Two sample t was used to compare continuous variables when appropriate. Fisher’s exact test was used to compare categorical variables. The level of significance was set at *p* < 0.05.

## Figures and Tables

**Table 1 gels-09-00758-t001:** Propriety and uses of antibiotic loaded hydrogels (ALH) and calcium sulphate (CS) beads. FRI: fracture related infections; OSE/TSE: one-/two-stage exchange; SSI: Surgical Site Infections; PJI: periprosthetic joint infection.

Antimicrobial Carrier	ALH	CS Beads
Composition	Hyaluronic acid and Poly-lactic acid	Calcium sulphate
Method of preparation in operating room	Add chosen antibiotic (liquid and/or powder) and dilute according to proportion until gel consistency is obtained	Add chosen antibiotic (liquid and/or powder) and dilute according to proportion and wait for solidification
Mode of application	Directly applied on plates, screws, nails, or prosthesis surfaces	Placed into bone cavities, intraarticular or subfascial
Duration of antibiotic release	24–72 h	Up to 30 days
Intended use by manufacturer	Prevention of infection in fractures and joint replacement	To fill voids, defects, and gaps caused by surgery, cysts, tumours, osteomyelitis, and traumatic injury
Described use by literature	Prevention of infection in fractures and joints replacement Reconstruction surgery for FRI OSE/TSE with cementless implant for PJI	OSE/TSE for PJI SSI Diabetic Foot Ulceration

**Table 2 gels-09-00758-t002:** Data about demographics features, localization, aetiology, and micro-organisms. CCI: commodity channel index; ASA risk score: American society of Anesthesiologists; *CoNS*: *coagulase negative staphylococci*.

	Total	DAPRI	DACRI	*p*-Value
Years (mean ± SD)	67.31 ± 10.79	66.86 ± 7.69	67.67 ± 12.18	0.8876
Sex				0.6143
Male	8 (50%)	4 (57.15%)	4 (44.44%)	
Female	8 (50%)	3 (42.85%)	5 (55.56%)	
CCI (mean ± SD)	3.06 ± 2.77	3.43 ± 3.69	2.78 ± 1.98	0.6572
ASA risk score (mean ± SD)	2.25 ± 0.45	2.14 ± 0.37	2.33 ± 0.50	0.4180
Localization				0.6143
Knee	8 (50%)	4 (57.15%)	4 (44.44%)	
Hip	7 (43.75%)	3 (42.85%)	4 (44.44%)	
Shoulder	1 (6.25%)	0 (0%)	1 (11.12%)	
Aetiology				0.6143
Hematogenous	8 (50%)	4 (57.15%)	4 (44.44%)	
Postoperative	8 (50%)	3 (42.85%)	5 (55.56%)	
Days to diagnosis (mean ± SD)	22.13 ± 9.08	23.86 ± 12.37	20.78 ± 5.93	
Micro-organisms				
*S. aureus*	7 (43.75%)	3 (33.34%)	4 (28.58%)	
*CoNS*	2 (12.50%)	1 (11.11%)	1 (7.14%)	
*E. coli*	3 (18.75%)	1 (11.11%)	2 (14.29%)	
*P. aeruginosa*	3 (18,75%)	1 (11.11%)	1 (7.14%)	
*A. baumanii*	1 (6.25%)	1 (11.11%)	0 (0%)	
*S. dysgalactiae*	1 (6.25%)	0 (0%)	1 (7.14%)	
*C. albicans*	1 (6.25%)	0 (0%)	1 (7.14%)	
Others	1 (6.25%)	0 (0%)	1 (7.14%)	
Polymicrobial	4 (25%)	1 (11.11%)	3 (21.43%)	
Culture negative	1 (6.25%)	1 (11.11%)	0 (0%)	

**Table 3 gels-09-00758-t003:** Intraoperative and postoperative data. CRP: C-reactive protein.

	Total	DAPRI	DACRI	*p*-Value
**Local Antibiotics**				
Vancomycin	2 (12.50%)	0 (0%)	2 (22.22%)	
Gentamicin + Vancomycin	10 (62.50%)	6 (85.71%)	4 (44.45%)	
Meropenem + Vancomycin	3 (18.75%)	1 (14.29%)	2 (22.22%)	
Fluconazole	1 (6.25%)	0 (0%)	1 (11,11%)	
**Outcomes**				
Infection controlled	15 (93.75%)	7 (100%)	8 (88.89%)	0.3624
Septic revision	1 (6.25%)	0 (0%)	1 (11.11%)	
CRP at last f-u (mg/L, mean ± SD)	3.75 ± 3.73	4.99 ± 4.62	2.81 ± 2.91	0.2680
Follow-up duration (mean ± SD)	26.12 ± 22.18	7.71 ± 1.80	40.44 ± 18.8	0.0007

**Table 4 gels-09-00758-t004:** Data about the whole population in our sample. ASA risk score: American Society of Anesthesiologists; ATB I.V.: intravenous antibiotic; ATB os: oral antibiotic; CRP: C-reactive protein; F-U: follow-up; *MSSA*: *methicillin-sensitive S. aureus*; *MRSA*: *methicillin-resistant S. aureus*; *CoNS*: *coagulase negative-staphylococci*; Pip/Tazo: piperacillin/tazobactam; Amoxi/Clav: amoxicillin-clavulanate; Sulb/Amp: sulbactam ampicillin.

	Sex	Age (Years)	ASA	Localization	Postoperative/ Hematogenous	Days from Symptoms Onset/Surgery	Microbiology	Surgical Treatment	Local Antibiotic 1	Local Antibiotic 2	Length of Stay (Days)	ATB I.V. (Active Ingredients)	ATB os (Active Ingredients)	Infection Free at last F-U (Y/N)	CRP last F-U (mg/L)	F-U (Months)
1	F	79	2	knee	haematogenous	26	MSSA	DACRI	Gentamycin	Vancomycin	19	Rifampicin, Amikacin, Fosfomycin	Dalbavancin, Amoxi/Clav	Y	5.90	17
2	M	40	2	hip	postoperative	13	MRSA, Gram -	DACRI	Gentamycin	Vancomycin	47	Daptomycin, Meropenem, Colistin	Ciprofloxacin	Y	4.36	16
3	M	64	2	knee	haematogenous	17	CoNS, Streptococcus dysgalactiae	DACRI	Vancomycin	-	11	Ceftriaxone, Linezolid, Pip/Tazo	Amoxi/Clav	Y	7.60	35
4	F	65	2	shoulder	postoperative	23	MSSA	DACRI	Gentamycin	Vancomycin	9	Pip/Tazo, Daptomycin, Dalbavancin	Rifampicin	Y	1.40	31
5	F	76	3	knee	haematogenous	26	MSSA	DACRI	Gentamycin	Vancomycin	23	Daptomycin, Cefazolin	Rifampicin, Levofloxacin	Y	0.00	28
6	F	78	3	hip	postoperative	12	Gram -, others	DACRI	Vancomycin	Meropenem	36	Targosid, Meropenem, Tazocin	Ciprofloxacin, Amoxi/Clav	Y	0.14	53
7	F	82	3	knee	haematogenous	18	Gram -	DACRI	Vancomycin	Meropenem	9	Ertapenem, Dalbavancin	Levoxacin, Doxycicline	Y	0.00	49
8	M	58	2	hip	postoperative	24	Gram -	DACRI	Vancomycin	-	8	Vancomycin, Rifampicin	Rifampicin, Amoxi/Clav	Y	2.80	64
9	M	67	2	knee	postoperative	28	others	DACRI	Fluconazole	-	4	-	Fluconazole	N	4.70	71
10	F	77	2	hip	postoperative	42	Colture negative	DAPRI	Gentamycin	Vancomycin	15	Cefazolin	Rifampicin, Levofloxacin	Y	5.00	7
11	M	67	3	hip	postoperative	25	Gram -	DAPRI	Gentamycin	Vancomycin	35	Meropenem, Daptomycin, Gentamicin, Ceftriaxone	Ceftriaxone	Y	3.30	11
12	M	65	2	hip	postoperative	30	MSSA, others	DAPRI	Gentamycin	Vancomycin	91	Daptomycin, Cefepime, Cefiderocol, Daptomycin, Sulb/Amp, Colistin	-	Y	0.06	9
13	M	66	2	hip	haematogenous	34	Gram -	DAPRI	Gentamycin	Vancomycin	15	Meropenem	Ciprofloxacin	Y	2.20	8
14	F	70	2	knee	haematogenous	15	CoNS	DAPRI	Vancomycin	Meropenem	16	Daptomycin, Pip/Tazo	Amox/Clav	Y	13.40	7
15	M	52	2	knee	haematogenous	13	MSSA	DAPRI	Gentamycin	Vancomycin	11	Daptomycin, Ceftobiprole, Cefazolin	Rifampicin, Levofloxacin	Y	11.2	6
16	F	71	2	knee	haematogenous	8	MSSA	DAPRI	Gentamycin	Vancomycin	23	Daptomycin, Pip/Tazo	Rifampicin, Levofloxacin	Y	6.00	6

## Data Availability

The data presented in this study are available upon request from the corresponding author.

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
