# Peer review of "Hydrogel Coating versus Calcium Sulphate Beads as a Local Antibiotic Carrier for Debridement Procedures in Acute Periprosthetic Joint Infection: A Preliminary Study"

_gels, 2023, doi:10.3390/gels9090758_

Round 1
Reviewer 1 Report
The manuscript entitled “Hydrogel coating versus calcium sulphate beads as local antibiotic carrier for debridement procedures in acute periprosthetic joint infection: a preliminary study” was aimed to demonstrate the debridement procedures and antibacterial action of hydrogel coating and calcium sulphate beads. However, throughout the full text, there are no enough antibacterial properties and debridement data to support efficiently the conclusion. Hence, I was not able to support its publication in Gels.
-
The paper language needs to be polished and improved further.
Author Response
The authors thank the reviewer for his feedback. We tried to improve the text based by reviewers suggestion.
Reviewer 2 Report
This article examines a new treatment option and its implications for DAIR surgery as a treatment for acute joint infection (PJI) by comparing the efficacy and safety of two different modified surgeries, DAPRI and DACRI. This reviewer has questions about this paper regarding the following:
(1) Contrary to the authors' goals, the results of hydrogel (ALH) and calcium sulfate (CS) beads used to release antibiotics immediately after surgery were not significantly different over the range of cases studied. Several factors need to be considered in this study. First, in retrospective studies, there are concerns about the accuracy of missing data or historical records. Incomplete or misinterpreted data can undermine the reliability of the results. Second, although basic t-tests and chi-squared tests were used in this study, these are not always the optimal statistical methods and have limitations, particularly in adequately controlling for small sample sizes and multivariate effects. The introduction of multivariate analysis should be considered, but multivariate analysis generally requires at least 10 to 20 cases per independent variable, and the number of cases in this study would be too small. Despite these shortcomings, the purpose and benefit to the reader of publishing this as a paper is not clear.
(2) This reviewer has concerns about whether this paper is consistent with the aims and scope of this journal. The journal Gel is intended for readers interested in gel materials. However, the article is a case-control report in medicine, and the chemistry of the hydrogels used in the treatment is not clear, nor is the research designed to develop gel materials.
Author Response
(1) Contrary to the authors' goals, the results of hydrogel (ALH) and calcium sulfate (CS) beads used to release antibiotics immediately after surgery were not significantly different over the range of cases studied. Several factors need to be considered in this study. First, in retrospective studies, there are concerns about the accuracy of missing data or historical records. Incomplete or misinterpreted data can undermine the reliability of the results. Second, although basic t-tests and chi-squared tests were used in this study, these are not always the optimal statistical methods and have limitations, particularly in adequately controlling for small sample sizes and multivariate effects. The introduction of multivariate analysis should be considered, but multivariate analysis generally requires at least 10 to 20 cases per independent variable, and the number of cases in this study would be too small. Despite these shortcomings, the purpose and benefit to the reader of publishing this as a paper is not clear.
The authors thank the reviewer for his feedback. We tried to improve the text keeping in mind this feedback; nevertheless, the several limitations are very well stated in the manuscript. Moreover the low number of cases is explained by the fact that this is a preliminary study over a relatively rare but devastating complication of joint replacement surgery. Moreover, there is a scarcity of literature about local readsobable carrier or coating in acute PJI. We hope that explanation, accompanied by all the implementation in the text we made, could be sufficient.
(2) This reviewer has concerns about whether this paper is consistent with the aims and scope of this journal. The journal Gel is intended for readers interested in gel materials. However, the article is a case-control report in medicine, and the chemistry of the hydrogels used in the treatment is not clear, nor is the research designed to develop gel materials.
The authors thank the reviewer for his feedback. We tried to improve the text keeping in mind this feedback trying to give a stronger link between the topic and the Journal, in which we believe that an hydrogel medical application, never published or described before for local carrier in acute PJI, is one of the most adequate recipient.
Reviewer 3 Report
Authors present preliminary study of use of hydrogel coating vs calcium sulphate beads for antibiotic carrier system for periprosthetic join infection (PJI). I believe that this is important problem and the presented results can be published after minor revision.
- the Introduction part have to be revised, Authors present 19 papers but the problem (PJI), the use of hydrogel and use calcium sulphate beads is quite rich.
- my second remark is the conclusion: I think it should be additional section at the end of the article where Authors could elaborate more on the results and further plans (after all, these are "preliminary study")
The language is fine.
Author Response
Authors present preliminary study of use of hydrogel coating vs calcium sulphate beads for antibiotic carrier system for periprosthetic join infection (PJI). I believe that this is important problem and the presented results can be published after minor revision.
The authors thank the reviewer for the feedback
- the Introduction part have to be revised, Authors present 19 papers but the problem (PJI), the use of hydrogel and use calcium sulphate beads is quite rich.
The authors thank the reviewer for the suggestion. Introduction has been improved according to reviewer’s suggestion.
- my second remark is the conclusion: I think it should be additional section at the end of the article where Authors could elaborate more on the results and further plans (after all, these are "preliminary study")
The authors thank the reviewer for the suggestion. Conclusion paragraph is now improved.
Reviewer 4 Report
Paper entitled: “Hydrogel coating versus calcium sulphate beads as local antibiotic carrier for debridement procedures in acute periprosthetic joint infection: a preliminary study” by Meo et al. Describes the difference between the use of antibiotic loaded hydrogel (ALH) and calcium sulphate (CS) beads in DAIR procedure. The manuscript needs major revision before being acceptance to published in gels journal
1- Lines 24, Table 1, the scientific names must be in italic, please check and revise throughout the manuscript.
2- In line 22, the abbreviation should be mentioned completely the first time, please check, and revised throughout the manuscript.
3- The authors should add a clear hypothesis at the end of the introduction.
4- Line 76, “67,3 ± 10,8” should be “67.3 ± 10.8”, please revise throughout the manuscript.
5- The total number of cases “16 cases” is low to achieve this study, please clarify.
6- Please add a “footnote” after each table to define the present abbreviation in this table.
7- What is the meaning of “CoNs” in Table 1.
8- Please clarify who to isolate and identify the pathogenic microbes.
9- Material and method section should be discussed in detail.
10- The manuscript needs deep discussion.
11- The authors should be adding a conclusion containing overall conclusion based on obtained results.
Some English expressions in the text needs improvement
Author Response
1- Lines 24, Table 1, the scientific names must be in italic, please check and revise throughout the manuscript.
The authors thank the reviewer for the suggestion. The text has been modified accordingly.
2- In line 22, the abbreviation should be mentioned completely the first time, please check, and revised throughout the manuscript.
The authors thank the reviewer for the suggestion. The text has been modified accordingly.
3- The authors should add a clear hypothesis at the end of the introduction.
The authors thank the reviewer for the suggestion. The text has been modified accordingly.
4- Line 76, “67,3 ± 10,8” should be “67.3 ± 10.8”, please revise throughout the manuscript.
The authors thank the reviewer for the suggestion. The text has been modified accordingly.
5- The total number of cases “16 cases” is low to achieve this study, please clarify.
The authors thank the reviewer for the suggestion. The text has been modified accordingly.
6- Please add a “footnote” after each table to define the present abbreviation in this table.
The authors thank the reviewer for the suggestion. The footnotes are already present.
7- What is the meaning of “CoNs” in Table 1.
The authors thank the reviewer for the suggestion. The meaning of CoNs is already written in footnotes.
8- Please clarify who to isolate and identify the pathogenic microbes.
The authors thank the reviewer for the suggestion. The methods section has been modified accordingly.
9- Material and method section should be discussed in detail.
The authors thank the reviewer for the suggestion. The methods and discussion section has been modified accordingly.
10- The manuscript needs deep discussion.
The authors thank the reviewer for the suggestion. The discussion section has been deepened with more literature about DAIR techniques.
11- The authors should be adding a conclusion containing overall conclusion based on obtained results.
The authors thank the reviewer for the suggestion. The conclusion section has been created and implemented accordingly to reviewer’s comments.
Round 2
Reviewer 1 Report
This article may be accepted
Author Response
Thank you for your quick answer
Reviewer 2 Report
Since I did not receive a fully satisfactory response to the points raised in the previous peer review, I would like to suggest improvements to the paper from a different perspective: Gels is intended for readers interested in gel materials. The authors should add a description and figures describing the physical property data of antibiotic loaded hydrogels (ALH) and calcium sulfate (CS) beads.Author Response
Since I did not receive a fully satisfactory response to the points raised in the previous peer review, I would like to suggest improvements to the paper from a different perspective: Gels is intended for readers interested in gel materials. The authors should add a description and figures describing the physical property data of antibiotic loaded hydrogels (ALH) and calcium sulfate (CS) beads.
Response: We’re sorry about reviewer’s complete satisfaction and we thanks him/her for the new perspective. A table with differences in composition and use was added in the text according to reviewer suggestion.

Reviewer 4 Report
The authors modify the manuscript accordingly but there are minor issue should be corrected before acceptance for publication.
1- The number in tables and text should be revised. For instance, (57, 15%) should be (57.15%).
2- The number of samples are low to achieve this analysis.
3- The conclusion should not contain references.
4- The isolation and identification procedures for microbial strain should be written in details
The English style and editing are satisfied
Author Response
1- The number in tables and text should be revised. For instance, (57, 15%) should be (57.15%).
The authors thank the reviewer for the suggestion. Changes were made in the text according to reviewer.
2- The number of samples are low to achieve this analysis.
The authors thank the reviewer for the comment. We’ve used Fisher exact test to compare categorical variables, and t-student test for continuous variables. There was an error in Methods statistical analysis description, that now has been corrected.
3- The conclusion should not contain references.
The authors thank the reviewer for the suggestion. Changes were made in the text according to reviewer.
4- The isolation and identification procedures for microbial strain should be written in details
The authors thank the reviewer for the suggestion. We have added a description of isolation and identification procedures for microbial strain in the materials and methods section
